# Multiple Blockholders and Firm Value: A Simulation Analysis

Annalisa Russino 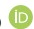

Department of Economics, Business and Statistics, University of Palermo, Viale delle Scienze Ed. 13,
90128 Palermo, Italy; annalisa.russino@unipa.it

**Abstract:** In this paper, we analyse the relationship between the distribution of ownership and firm value in the presence of multiple blockholders. In recent years, the topic has attracted the attention of many scholars. Yet, the empirical evidence on the relationship between the distribution of ownership among large shareholders and firm value has been non-conclusive and contradictory. We focus on the interaction between a controlling block of shareholders and a non-controlling block that can monitor the largest controlling block. We develop and simulate a simple model combining the two effects related to the presence of additional blockholders that can monitor the largest controlling block of shareholders. The first concerns the incentives of the controlling blockholders to expropriate other shareholders (the alignment effect); the second concerns the incentives for non-controlling blockholders to exercise monitoring activities (the monitoring effect). We examine the influence of the distribution of ownership between controlling and non-controlling shareholders on the total amount of company resources diverted to provide private benefits to controlling shareholders. Since net firm value is decreasing in the amount of company resources diverted, our analysis sheds light on the relationship between the ownership structure and firm value. We show that, in the presence of multiple blockholders, the relationship between ownership concentration and firm value may change depending on the relative size of the shareholders and the relative size of private benefits of control. Our results help in understanding the variety of shapes that have been empirically detected, and shed some light on the conditions that make optimal diversions, as a function of the level of ownership concentration, monotone (increasing and decreasing) or non-monotone.

**Keywords:** corporate governance; ownership structure; multiple blockholders; principal–principal conflicts; valuation

## 1. Introduction

Much of the corporate governance literature on the relationship between ownership structure and firm value has focused on two extreme cases: (1) dispersed ownership structures with atomistic shareholders, and (2) structures with one large, controlling, owner combined with atomistic shareholders (see Laeven and Levine (2008)). It is only recently that scholars have started to look at intermediate ownership configurations with multiple large shareholders.

A fundamental feature of ownership structures with large shareholders is that shareholders are typically knowledgeable about firm operations and implicated in its management. Thus, concentrating shareholdings addresses the familiar agency problem arising from the separation of ownership and control. As a consequence, ownership structures with large shareholders can be effective internal corporate governance mechanisms alternative to external governance devices such as a well-functioning market for corporate control. The key governance conflict in such situations is the abuse of power by controlling shareholders at the expenses of minority shareholders. That is, a conflict may arise between controlling shareholders and non-controlling shareholders over the extraction of private benefits of control (Shleifer and Vishny 1997; Gilson 2006). Following Gilson (2006) taxonomy, in this paper we focus our attention on inefficient controlling shareholder structures defined as situations where non-controlling shareholders are net worse off.

Theoretical research suggests that multiple large shareholders can be effective in mitigating the conflict of interests between controlling and non-controlling shareholders. In particular, scholars have suggested three ways in which multiple blockholders can positively affect firm value. The first channel is the process of coalition formation to ensure control of the firm (Bennedsen and Wolfenzon 2000; Gomes and Novaes 2005). The second channel is the threat of exiting through trading (Edmans and Manso 2011). Finally, the third channel highlights the role of non-controlling blockholders as monitors of controlling blockholders (Pagano and Roell 1998; Bloch and Hege 2003; Dhillon and Rossetto 2015).

Despite the positive effect of multiple blockholders on firm value indicated by the theoretical research, the empirical evidence on the relationship between the distribution of ownership in the presence of large shareholders and firm value has been inconclusive and contradictory. In particular, the empirically detected relationship between the measures of blockholders' ownership concentration and measures of firm value has shown a variety of forms: monotone decreasing (Maury and Pajuste 2005; Laeven and Levine 2008; Attig et al. 2009), monotone increasing (Konijn et al. 2011), non-monotone with an inverted *U*-shape (Thomsen and Pedersen 2000; De Miguel et al. 2004; Cai et al. 2016), and non-monotone with a *U*-shape (Gutiérrez et al. 2012; Nagar et al. 2011; Lozano et al. 2016; Basu et al. 2017; Russino et al. 2019; Kong et al. 2020).

In this paper, we focus on the monitoring role of non-controlling blockholders. In particular, we develop and simulate a simple model combining the two effects related to the presence of additional blockholders that can monitor the largest controlling block of shareholders: the alignment and monitoring effects. The two effects arise because the distribution of ownership among the controlling and non-controlling block of shareholders, on one side, affects the incentives of the controlling blockholders to expropriate other shareholders (the alignment effect), and, on the other side, the incentives of non-controlling blockholders to exercise monitoring activities (the monitoring effect).

We analyse the relationship between the distribution of ownership between controlling and non-controlling shareholders and the total amount of company resources diverted to provide private benefits to the controlling shareholders. Since net firm value is decreasing in the amount of company resources diverted, our analysis sheds light on the relationship between the ownership structure and firm value.

The most recent empirical evidence has stressed the importance of three main factors affecting the relationship between the ownership structure and firm value. First, the framework of the analysis: if the selected sample of firms is characterized by highly concentrated or more dispersed ownership structures (as it happens in samples of large and/or listed firms versus samples of small and/or unlisted firms). Second, the identities of the blockholders. Third, the interaction between blockholders driven by their relative ownership stakes (see Lozano et al. 2016; Basu et al. 2017; Russino et al. 2019; Kong et al. 2020). However, implementing adequate empirical tests capturing these factors is quite challenging. For instance, it is hard to distinguish when the largest shareholder has full control on a firm's decisions besides the rough majority rule. Even more difficult to analyse is the influence of the interaction between blockholders and the importance of their relative stakes. Focusing on the aggregate sum of the stakes of blockholders (as, for instance, in Kong et al. (2020)) neglects the influence of their relative sizes. Alternatively, concentrating the attention on the difference in ownership stakes of the largest and lower ranked blockholders (as in Maury and Pajuste (2005) and Laeven and Levine (2008)) does not take into account the influence of the level of the stakes. Finally, focusing on the stake size of the largest shareholder combined with dummies indicating the presence (and/or a cut-off size) of additional blockholders (as in Lozano et al. (2016); Basu et al. (2017)) is a rough and imprecise way of analysing the impact of the blockholders' relative size.

The main contribution of this paper is to provide a simple model where the interaction between a controlling and a non-controlling block of shareholders driven by their sizes can be analysed, and the consequent relationship between the distribution of ownership between the two blocks and firm value can be detected. We show that in the presence of

multiple blockholders the relationship between ownership concentration and firm value may change depending on the sizes of the controlling and non-controlling shareholders and the size of private benefits of control[1]. The remainder of this paper is structured as follows. Section 2 describes the model. Section 3 describes the setting used for the simulations and discusses the results. Section 4 concludes our work.

## 2. The Model

To make the analysis simpler, we consider a setting where there are only two blocks[2]: a block of shares big enough to allow the shareholders to control the firm operations, and a second smaller block of shares that may induce the shareholders to monitor the decisions taken by the controlling block[3]. The two blocks can be either individual shareholders or the result of coalitions of shareholders agreeing to act in unison[4].

Let $SH_1$ and $SH_2$, respectively, be the percentage ownership of the controlling block and the non-controlling block. We assume that besides the two largest blocks there is a residual percentage ownership ($R$) in the hands of atomistic or passive shareholders. In this framework, given the level of residual diffused ownership $R$, ownership concentration can be measured by the size of the fractional ownership of the controlling block. Let $V$ be the firm value and $D \in [0, \overline{D}]$, with $\overline{D} < V$, being the amount of corporate resources that the controlling block can divert to enjoy private benefits of control. The diversion of corporate resources represents a value reduction to the firm that will yield private benefits to the controlling shareholder. Thus, we posit that, for each unit of corporate resources diverted, the controlling shareholder will get $B < 1$ private benefits. That is, each unit diverted will generate a value loss to the firm, and $(1 - B)$ is a measure of the inefficiency of the extraction of private benefits of control[5]. We suppose that the diversion of corporate resources is an activity that requires some effort ($e$). The non-controlling blockholder can keep down the amount of diversion through monitoring ($m$). The total amount of resources diverted will depend on the choices of $e$ and $m$. For simplicity, we assume that the amount of corporate resources diverted is an additive function: that is $D(e, m) = D_e + D_m$. We assume that $D_e$ is an increasing concave function. That is, the total amount of corporate resources diverted will increase as the effort implemented by the controlling shareholder increases ($D'_e > 0$), but the marginal positive effect of the increase in effort decreases ($D''_e < 0$). At the same time, we assume that $D_m$ is a decreasing convex function. Thus, the total amount of corporate resources diverted decreases in the level of monitoring chosen by the non-controlling shareholder ($D'_m < 0$), and the marginal negative effect on an increase in $m$ is decreasing ($D''_m > 0$). As conventional, we assume convex cost functions for both the controlling and non-controlling blockholders ($C'_e > 0, C''_e > 0$ and $C'_m > 0, C''_m > 0$, respectively).

We model the choice of $e$ and $m$ as non-observable actions; therefore, the two blockholders will simultaneously choose $e$ and $m$ to maximize their utility functions. Assuming utility functions linear in wealth, the objective functions of the two blockholders can be written as follows,

$$
\begin{aligned}
U_1 &= SH_1(V - D(e, m)) + BD(e, m) - C_e \\
U_2 &= SH_2(V - D(e, m)) - C_m
\end{aligned}
$$

The interior solutions, the optimal $(e^*, m^*)$, are the solutions of the first-order conditions,

$$
\begin{aligned}
e^* : U'_1 &= -SH_1 D'_e + BD'_e - C'_e = 0 \\
m^* : U'_2 &= -SH_2 D'_m - C'_m = 0
\end{aligned}
\tag{1}
$$

with second-order conditions given by,

$$
\begin{aligned}
(D''_e(B - SH_1) - C''_e) &< 0 \\
(-SH_2 D''_m - C''_m) = [-(1 - SH_1)D''_m - C''_m)] &< 0
\end{aligned}
$$

If the fractional ownership of the controlling blockholder is greater than $B$ $(SH_1 \geq B)$ diversion will not be profitable for the controlling shareholder $(e^* = 0)$[6]. The interior solutions correspond to the case where there are incentives to divert corporate resources or $(B - SH_1) > 0$. The total optimal amount of diversion $D^*$ is a function of the chosen levels of effort and monitoring $(e^*$ and $m^*)$. We are interested in analysing how a change in the distribution of ownership between the two blocks may affect the optimal amount of diversion. Since, for a given $R$, a larger share of ownership of the controlling block corresponds to a more concentrated ownership structure, we can look at the impact of increasing ownership concentration on $D^*$, analysing the function $D^*(SH_1)$.

The effect of an increase in the percentage ownership of the controlling blockholder $(SH_1)$ on $D^*$ is given by,

$$\frac{dD^*}{dSH_1} = D'_e \frac{de^*}{dSH_1} + D'_m \frac{dm^*}{dSH_1} \tag{2}$$

where $\frac{de^*}{dSH_1}$ and $\frac{dm^*}{dSH_1}$ are obtained totally differentiating the first-order conditions.

Totally differentiating system (1), where $SH_2 = (1 - SH_1 - R)$ and $R$ is the residual ownership share not in the hands of the two largest shareholders, we obtain[7],

$$-D'_e dSH_1 + de[(B - SH_1)D''_e - C''_e] = 0$$
$$D'_m dSH_1 + dm[-(1 - SH_1 - R)D''_m - C''_m] = 0$$

Considering $de, dm$ endogenous variables and $dSH_1$ as exogenous, we can solve the above system to obtain $\frac{de}{dSH_1}$ and $\frac{dm}{dSH_1}$. For interior solutions, satisfying the first- and second-order conditions, we obtain,

$$\frac{de^*}{dSH_1} = \frac{D'_e[(1 - SH_1 - R)D''_m + C''_m]}{[(B - SH_1)D''_e - C''_e][(1 - SH_1 - R)D''_m + C''_m]} < 0$$
$$\frac{dm^*}{dSH_1} = \frac{D'_m[(B - SH_1)D''_e - C''_e]}{[(B - SH_1)D''_e - C''_e][(1 - SH_1 - R)D''_m + C''_m]} < 0$$

Both the optimal values $e^*$ and $m^*$ decrease as the fraction of shares in the hands of the controlling shareholder increases.

It follows that the total effect of an increase in the size of the controlling block on the optimal amount of resources diverted, defined in Equation (2), is equal to the summation of the two terms with opposite signs,

$$\frac{dD^*}{dSH_1} = \underbrace{D'_e \frac{\partial e^*}{\partial SH_1}}_{<0} + \underbrace{D'_m \frac{\partial m^*}{\partial SH_1}}_{>0} \tag{3}$$

Equation (3) assumes a positive or negative sign depending on the relative size of its two components. The first represents the alignment effect: when the size of the fractional ownership of the largest blockholder increases they will bear a greater part of the value loss generated by the extraction of private benefits of control. This, in turn, will decrease the optimal level of effort devoted to the extraction of private benefits implying a lower level of corporate resources diverted. Thus, the increase in $SH_1$ makes the controlling blockholder's interests more aligned with firm value maximization. The second represents the monitoring effect: keeping fixed the fractional ownership of passive or atomistic shareholders, an increase in $SH_1$ implies a decrease in $SH_2$. When the percentage ownership of the non-controlling shareholder decreases, their incentives to monitor will also decrease. The optimal level of monitoring will be lower, and this in turn will imply an increase in the amount of corporate resources diverted.

The resulting total effect on $D^*$ of an increase in $SH_1$ will depend on the relative size of the alignment and monitoring effects. Since net firm value is decreasing in $D^*$, the optimal level of diversion at any given level of ownership concentration ($D^*(SH_1)$) will, correspondingly, determine the net firm value distributed between the blockholders.

In the next section we will provide simulations of the model and show that the combination of these two effects, at different levels of the parameters, may significantly change the relationship between ownership concentration and net firm value.

## 3. Simulations and Results

In this section, we specify the functional forms and calibrate parameters of the model discussed in the previous section to analyse, through simulations, the relationship between the optimal amount of corporate resources diverted and the distribution of ownership between the two blocks. We will focus on the impact of increasing the ownership concentration that, in our setting, is measured by the fractional ownership in the hands of the controlling blockholder.

We consider two blocks holding at least 10% of the firm shares. In Table 1, we summarize the variables used in the simulations specifying the range of values and constraints that they must satisfy.

**Table 1.** Variables description.

| Variable | Description | Range of Values | Constraints |
|---|---|---|---|
| $V$ | Firm value | $V \in [0,1]$ | |
| $D$ | Total amount of corporate resources diverted | $D \in [0, V)$ | $D^* < V$ |
| $SH_1$ | Percentage ownership of the largest blockholder | $SH_1 \in (0.10, (1 - SH_2 - R)]$ | $SH_1 > 10\%$ |
| $SH_2$ | Percentage ownership of the second largest blockholder | $SH_2 \in [0.10, (1 - SH_1 - R)]$ | $SH_1 > SH_2 \geq 10\%$ |
| $R$ | Residual diffused ownership | $R \in [0, (1 - SH_1 - SH_2)]$ | |
| $B$ | Per unit private benefits of control | $B \in (0.10, 1)$ | $B > SH_1$ |
| $e$ | Intensity of effort to extract private benefits of control | $e \in [0,1]$ | |
| $m$ | Intensity of monitoring | $m \in [0,1]$ | |

All the simulations are implemented using MatlabR2022a[8]. We solve the optimization problem of the two blockholders for each specification of the parameters and determine the optimal choices $(e^*, m^*)$ and resulting optimal level of corporate resources diverted $D^*$.

### 3.1. $D^*(SH_1)$ at Different Values of B Fixing R

We assume that the controlling blockholder has a comparative advantage relative to the non-controlling blockholder in affecting the firm's choices and activities[9]. In the calibration of the model the comparative advantage of the largest blockholder can be implemented through the definition of the functions describing the effect of the level of effort and monitoring on the total amount of corporate resources diverted and/or the blockholder's cost functions. For simplicity we specify all the functions as power functions,

$$
\begin{aligned}
D(e,m) &= D_e + D_m = e^a - m^b \\
C_e &= \frac{e^c}{2} \\
C_m &= \frac{m^d}{2}
\end{aligned}
$$

In this setting, the comparative advantage of the controlling blockholder can be implemented through the calibration of the parameters corresponding to the power of the functions $(a, b, c, d)$. The relative size of $(a, b)$ and $(c, d)$ will determine, respectively, the relative curvature of the diversion and cost functions. Setting $a < b$ implies that $e^a$ is more concave than $m^b$, while setting $c > d$ implies that the cost function of the controlling blockholder is more convex than the cost function of the second largest blockholder[10].

We start modelling the comparative advantage of the largest blockholder through the relative curvature of the diversion functions.

In Figure 1, we plot the simulated $D^*$ as a function of the percentage ownership of the largest blockholder ($SH_1$), at different levels of the per unit private benefits of control ($B$), fixing $a = 0.12 < b = 0.3$, $c = d = 2$, and $V = 1$. As a base case we assume that the percentage of ownership in the hands of atomistic shareholders ($R$) is very small or close to zero. This situation represents an extreme case of closely held firms[11], but it is not unrealistic. For instance, in a recent survey on the ownership structure and governance of Italian firms, Baltrunaite et al. (2019) showed that between 96% and 98% of the ownership is in the hands of the two largest shareholders. Accordingly, the top panel of Figure 1 shows the results obtained when the residual diffuse ownership is set equal to zero ($R = 0$). In this case the ranges of admissible values for the fractional ownership of the two blockholders are, respectively, $SH_1 \in (0.5, 0.9]$ and $SH_2 \in [0.10, 0.5)$ while $B \in (0.5, 1)$. The graphs show that the relationship between the optimal total amount of corporate resources diverted ($D^*$) and the size of the fractional ownership of the controlling blockholder ($SH_1$) changes as $B$ varies: moving from the highest plot, corresponding to the case where $B = 0.95$, to the lowest plot, obtained assuming $B = 0.6$, $D^*(SH_1)$ changes from monotonically increasing to inverted $U$-shaped and monotonically decreasing. At moderately high levels of the per unit private benefits of control, $D^*$ is a non-monotone function of $SH_1$. In particular, $D^*(SH_1)$ has an inverted $U$-shape: starting from the lowest admissible value of $SH_1$, an increase in $SH_1$ raises $D^*$ up to a threshold that is higher as $B$ increases. When $SH_1$ is relatively low, that is at low levels of ownership concentration, the alignment effect related to an increase in the size of the largest controlling blockholder will be lower than the monitoring effect and the resulting total optimal amount of resources diverted will increase. When a relatively high level of ownership concentration is reached, the alignment effect will prevail on the monitoring effect and $D^*$ will decrease in $SH_1$. The inverted $U$-shaped relationship is more pronounced as $B$ increases.

The bottom panel of Figure 1 shows what happens when $R$, the fraction of diffused ownership, is set equal to 0.20. The increase in $R$ shifts down the ranges of admissible values of the fractional ownership of the two blockholders ($SH_1 \in (0.4, 0.7]$ and $SH_2 \in [0.10, 0.4)$ while $B \in (0.4, 1)$). In this case the alignment effect will be much less effective and $D^*(SH_1)$ will be a monotone increasing function from high to moderate levels of $B$ ($B \geq 0.75$ in our simulations).

Next, we replicate the simulations making the comparative advantage of the controlling blockholder stronger adding a difference in the curvature of the cost functions. In Figure 2, we plot the simulated $D^*$ as a function of percentage ownership of the largest blockholder ($SH_1$) at different levels of private benefits of control ($B$), fixing $a = 0.12 < b = 0.3$, $c = 4 > d = 2$, and $V = 1$.

As before, the top panel shows $D^*(SH_1)$ at different levels of $B$ setting $R = 0$, while the bottom panel shows the results when $R = 0.20$. The general pattern does not change: when $R = 0$, $D^*(SH_1)$ has an inverted $U$-shape at moderately high levels of $B$ (from $B = 0.65$ to $B = 0.9$) that becomes more pronounced as $B$ increases. As before, the increase in $R$ makes the alignment effect much less effective and $D^*(SH_1)$ becomes a monotone increasing function from high to moderate levels of $B$ ($B \geq 0.75$). However, increasing the comparative advantage of the largest blockholder has an impact on the steepness of $D^*(SH_1)$: it generally makes $D^*$ more sensitive to changes in $SH_1$.

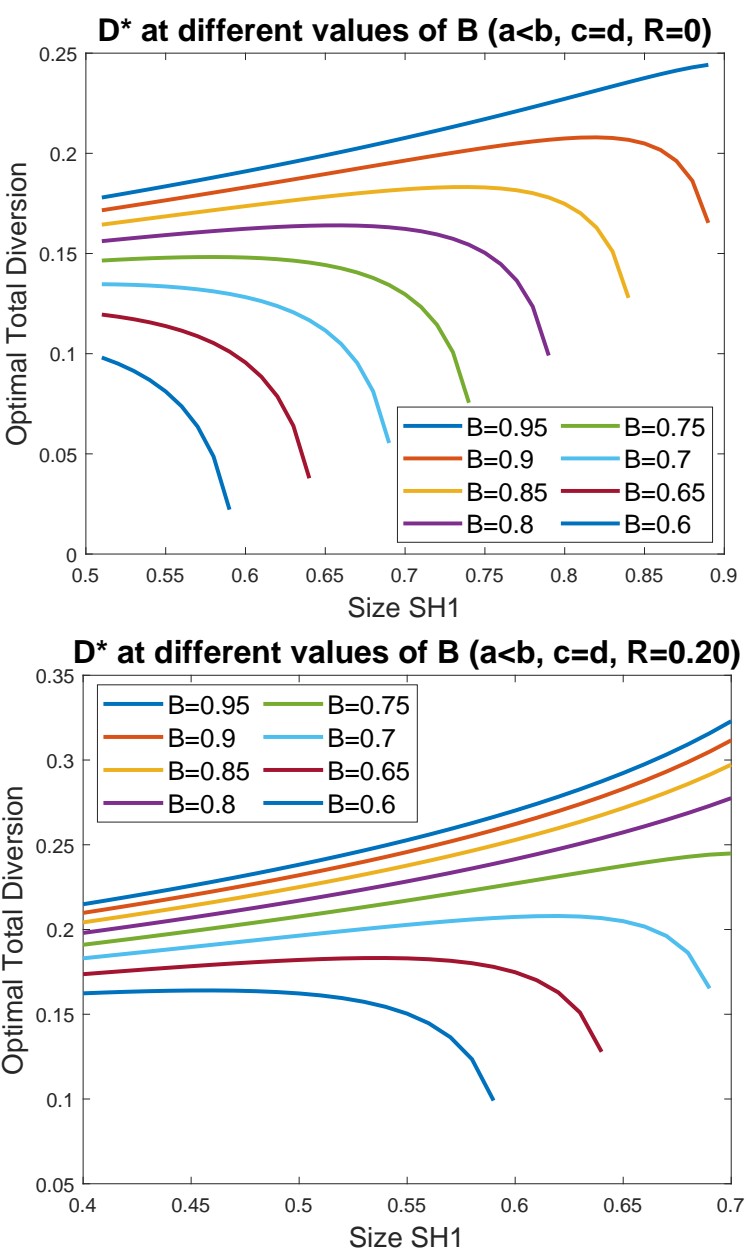

**Figure 1.** $D^*(SH_1)$ at two levels of $R$ ($a < b$ and $c = d$). In the two panels, the highest graph corresponds to the case where $B = 0.95$, and the lowest to the case where $B = 0.6$. The intermediate plots follow the order of the legend.

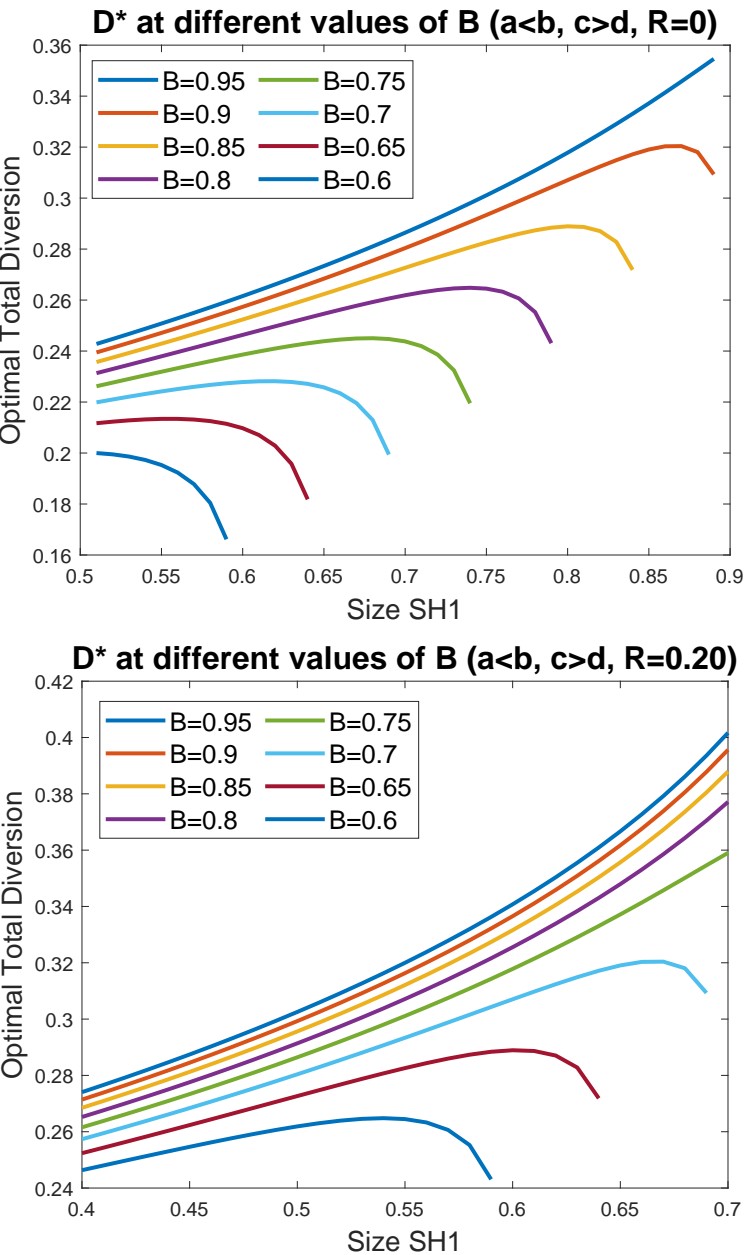

**Figure 2.** $D^*(SH_1)$ at two levels of $R$ ($a < b$ and $c > d$). In the two panels, the highest graph corresponds to the case where $B = 0.95$ and the lowest to the case where $B = 0.6$. The intermediate plots follow the order of the legend.

### 3.2. $D^*(SH_1)$ at Different Values of R Fixing B

Figures 3 and 4 show $D^*(SH_1)$ at different levels of $R$ fixing two levels of $B$. Increasing the fraction of diffused ownership shifts down the ranges of admissible values of the fractional ownership of the blockholders. For instance, setting $R = 0.20$ implies $SH_1 \in (0.4, 0.7]$ and $SH_2 \in [0.10, 0.4)$ while $R = 0.30$ implies $SH_1 \in (0.35, 0.6]$ and $SH_2 \in [0.10, 0.35)$. As a consequence, the alignment effect related to increases of $SH_1$ will be less effective, while the corresponding reduction in the optimal level of monitoring ($m^*$) will have a stronger effect on $D^*$.

In the top panel of Figure 3, $D^*(SH_1)$ becomes monotonically increasing as long as $R \geq 0.15$. The bottom panel shows a similar pattern but, given the low level of private benefits of control used in the simulations, $D^*(SH_1)$ shifts up as $R$ increases and changes shape from monotonically decreasing at $R = 0$ to inverted $U$-shaped as $R \geq 0.20$.

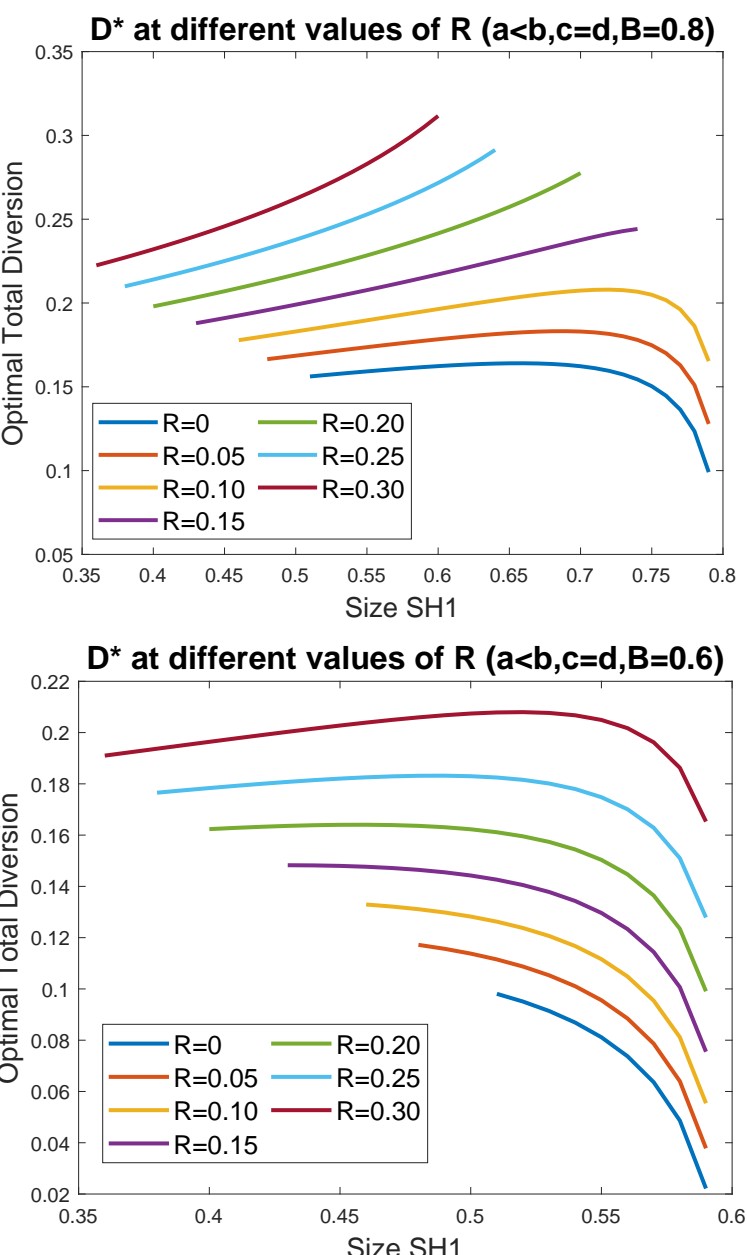

**Figure 3.** $D^*(SH_1)$ at $B = 0.6$ ($a < b$ and $c = d$). In the two panels, the highest graph corresponds to the case where $R = 0.30$ and the lowest to the case where $R = 0$. The intermediate plots follow the order of the legend.

In Figure 4, we replicate the simulations adding a difference in the convexity of the cost functions of the two blockholders. As before, we see a greater sensitivity of $D^*$ to changes in $SH_1$. In general, we notice an increase in the steepness of the simulated functions $D^*(SH_1)$.

In order to grasp the size of the impact of increasing $SH_1$ on $D^*$, in Table 2 we report the maximum value of $D^*$, and the corresponding level of $SH_1$, relative to three different levels of private benefits of control ($B$) and three levels of diffused ownership ($R$).

We can summarize the main findings of the analysis as follows. First, for each level of the per unit private benefits of control (the parameter $B$) increasing the fractional ownership in the hands of passive shareholders (i.e., increasing $R$) will shift down the range of admissible values for the stakes of both the controlling and non-controlling blockholders. The alignment effect related to increases in the stake of the controlling blockholder will tend to become relatively small compared to the concomitant reduction in the monitoring

incentives of the non-controlling block. Thus, the maximum optimal amount of resources diverted rises and, in general, the function $D^*(SH_1)$ will shift up. Second, for each level of diffused ownership ($R$), increasing the amount of private benefits of control per unit of resource diverted (increasing $B$) increases the maximum total amount of resources diverted. Third, the function $D^*(SH_1)$ changes as the parameters $B$ and $R$ vary.

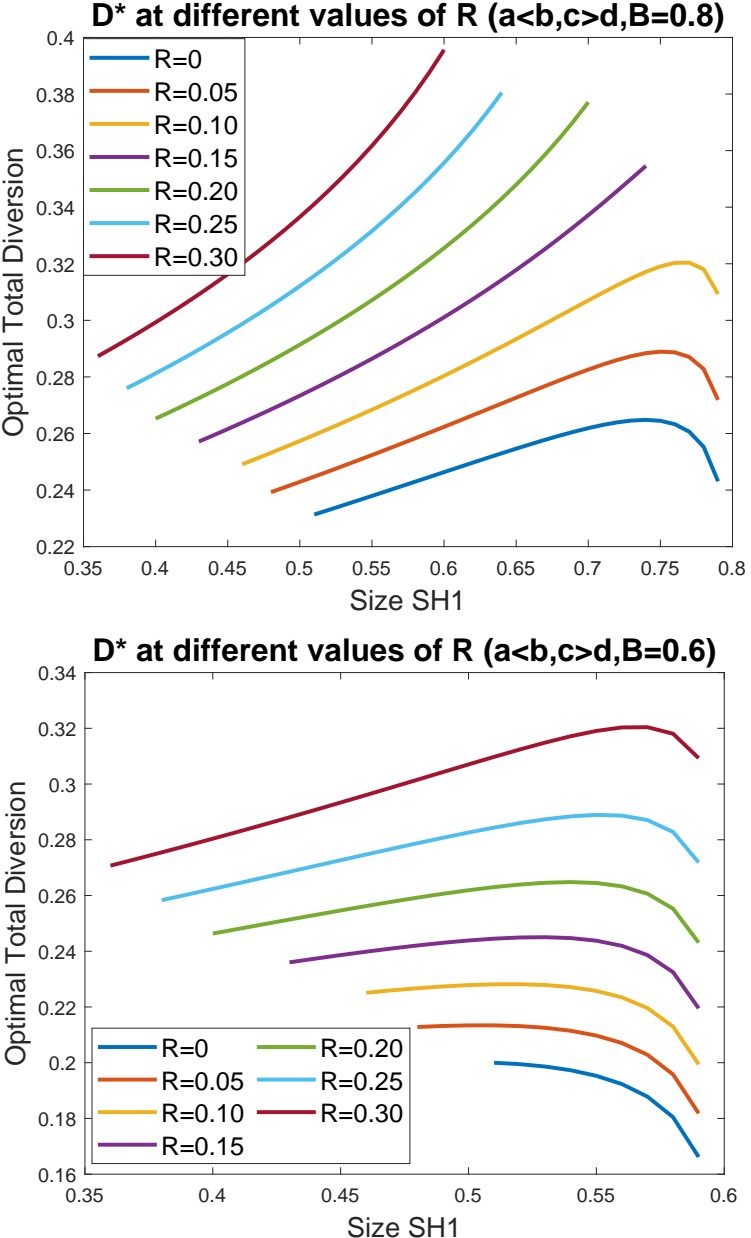

**Figure 4.** $D^*(SH_1)$ at two levels of $B$ ($a < b$ and $c > d$). In the two panels, the highest graph corresponds to the case where $R = 0.30$, and the lowest to the case where $R = 0$. The intermediate plots follow the order of the legend.

Our results concerning the role of $R$, the fractional ownership of atomistic share-holders, help understanding the contradictory empirical evidence observed when using samples of listed (and/or large) firms versus samples of unlisted (and/or small) firms[12]. Indeed an important difference between the two types of samples is the corresponding size of $R$. Typically, unlisted firms are closely held firms with few large blockholders holding illiquid shares and a very small fraction of atomistic shareholders. Therefore, these samples are settings better suited to detect the counterbalancing alignment and monitoring

effects, and where for sufficiently high levels of private benefits of control the relationship between ownership concentration and optimal diversion becomes an inverted *U*-shaped. Correspondingly, since net firm value is decreasing in $D^*$, in these settings the relationship between ownership concentration and net firm value will be *U*-shaped.

**Table 2.** Max value of $D^*$ with the corresponding level of $SH_1$.

| **First Case: $a < b$ and $c = d$** | | | |
|---|---|---|---|
| $R = 0$ | $B = 0.8$ | $B = 0.7$ | $B = 0.6$ |
| max $D^*$<br>$SH_1$<br>Shape $D^*(SH_1)$ | 0.1640<br>0.66<br>inverted U-shape<br>max: interior point | 0.1347<br>0.51<br>monotone decreasing<br>max: lowest extreme point | 0.0981<br>0.51<br>monotone decreasing<br>max: lowest extreme point |
| $R = 0.20$ | $B = 0.8$ | $B = 0.7$ | $B = 0.6$ |
| max $D^*$<br>$SH_1$<br>Shape $D^*(SH_1)$ | 0.2775<br>0.7<br>monotone increasing<br>max: highest extreme point | 0.2080<br>0.62<br>inverted U-shape<br>max: interior point | 0.1640<br>0.46<br>inverted U-shape<br>max: interior point |
| $R = 0.30$ | $B = 0.8$ | $B = 0.7$ | $B = 0.6$ |
| max $D^*$<br>$SH_1$<br>Shape $D^*(SH_1)$ | 0.3116<br>0.6<br>monotone increasing<br>max: highest extreme point | 0.2775<br>0.6<br>monotone increasing<br>max: highest extreme point | 0.2080<br>0.52<br>inverted U-shape<br>max: interior point |
| **Second Case: $a < b$ and $c > d$** | | | |
| $R = 0$ | $B = 0.8$ | $B = 0.7$ | $B = 0.6$ |
| max $D^*$<br>$SH_1$<br>Shape $D^*(SH_1)$ | 0.2648<br>0.74<br>inverted U-shape<br>max: interior point | 0.2282<br>0.62<br>inverted U-shape<br>max: interior point | 0.20<br>0.51<br>monotone decreasing<br>max: lowest extreme point |
| $R = 0.20$ | $B = 0.8$ | $B = 0.7$ | $B = 0.6$ |
| $D^*$<br>$SH_1$<br>Shape $D^*(SH_1)$ | 0.3771<br>0.7<br>monotone increasing<br>max: highest extreme point | 0.3204<br>0.67<br>inverted U-shape<br>max: interior point | 0.2648<br>0.54<br>inverted U-shape<br>max: interior point |
| $R = 0.30$ | $B = 0.8$ | $B = 0.7$ | $B = 0.6$ |
| max $D^*$<br>$SH_1$<br>Shape $D^*(SH_1)$ | 0.3956<br>0.6<br>monotone increasing<br>max: highest extreme point | 0.3771<br>0.6<br>monotone increasing<br>max: highest extreme point | 0.3204<br>0.57<br>inverted U-shape<br>max: interior point |

## 4. Conclusions

We have shown that the combination of the alignment and monitoring effects may change the shape of the relationship between the optimal amount of resources diverted and ownership concentration. Our results shed some light on the conditions that make the optimal diversion, as a function of the level of ownership concentration, monotone (increasing and decreasing) or non-monotone.

In particular, at extreme values of the parameter controlling the level of private benefits of control (*B*) the optimal amount of diversion will be a monotone function of the fractional ownership of the controlling blockholder: precisely, a monotone increasing function at the highest levels of *B*, and a monotone decreasing function at the lowest levels of *B*. At

intermediate levels of $B$ the optimal amount of diversion as a function of $SH_1$ will have an inverted $U$-shape. That is, when $SH_1$ increases around its minimum value the reduction in the optimal amount of diversion due to the alignment effect will be more than offset by the increase in $D^*$ due to the concomitant reduction in the optimal level of monitoring. The slope of $D^*(SH_1)$ is positive when $SH_1$ is small and becomes negative when $SH_1$ is big. Thus, an increase in $SH_1$ raises the optimal amount of diversion when $SH_1$ is small, whereas it decreases the optimal amount of diversion when $SH_1$ is large. Given that net firm value is decreasing in the amount of corporate resources diverted, we obtain a $U$-shaped relationship between net firm value and ownership concentration represented, in our setting, by the size of $SH_1$. Starting from its minimum level, an increase in $SH_1$ will have a negative effect on net firm value at relatively low levels of $SH_1$ and a positive effect at relatively high levels of $SH_1$.

Additionally, we find that the greater is the comparative advantage of the controlling blockholder in influencing a firm's operations, the higher the sensitivity of $D^*$ to changes in $SH_1$, and thus the steepness of $D^*(SH_1)$.

Finally, we show that an increase in the fractional diffused ownership (represented by the parameter $R$) shifts down the range of admissible values of the controlling blockholder's stake. When a substantial fraction of the ownership is in the hands of atomistic shareholders the largest blockholder may have effective control of the firm with a relatively low fraction of ownership. In such situations, an increase in ownership concentration will increase the optimal level of diversion for a wider range of values of $B$. When private benefits of control are substantial, increasing the percentage of diffused ownership will make the relationship between ownership concentration and optimal diversion monotone increasing. Correspondingly, the relationship between ownership concentration and firm value will be monotone decreasing.

These findings help in understanding the variety of shapes of the relationship between the distribution of ownership among blockholders and firm value empirically detected. In particular, the model sketched in this paper is well-suited to describe the counterbalance of forces related to the distribution of ownership between blockholders in closely held companies. Such firms are characterized by highly concentrated ownership structures among few blockholders, involvement of the largest shareholder in the management of the firm, and illiquid shares. They represent settings where market-oriented corporate governance devices (such as hostile takeovers, product market competition, and market pressure through "exit" or trading a firm's shares) are not effective and where shareholders are left with a "voice", or direct intervention, to influence firm value. In this framework the ownership distribution will affect the principal–principal conflict between the controlling and non-controlling blocks.

In this paper, we show that shifting towards a more concentrated ownership structure (where the share of the controlling block increases at the expense of the share of the non-controlling block) will affect the optimal amount of company resources diverted differently depending on two parameters: the per unit amount of private benefits of control and the size of the share of diffused ownership. The different shapes are driven by the counterbalance between the alignment of interests of the controlling blockholder and the monitoring incentives of the non-controlling block.

Specifically, we show that the relationship between optimal total diversion and ownership concentration becomes a non-monotone inverted $U$-shaped at moderately high levels of the per unit private benefits of control and at relatively low levels of the share of diffused ownership. Correspondingly, in such cases the relationship between net firm value and ownership concentration will be $U$-shaped. This finding is consistent with the empirical evidence relative to frameworks where ownership is highly concentrated (see Nagar et al. (2011); Lozano et al. (2016); Russino et al. (2019)).

Additionally, consistent with the empirical evidence, we show that in situations where the ownership is more dispersed ($R$ increases), as typically happens in the case of large listed firms, the relationship between ownership concentration and net firm value will tend to be monotone decreasing (see, for example, Laeven and Levine (2008) and Lozano et al. (2016)).

Concerning the limitations of the paper, we acknowledge that we focus on principal–principal conflicts assuming that the familiar managerial agency problem is absent due to the presence of a large controlling blockholder. Our model does not explicitly identify the threshold size of ownership necessary to guarantee convergence of interests between the management and shareholders. The empirical evidence on managerial ownership shows that increasing ownership will have a positive effect on firm value up to a threshold value and a negative effect afterwards (see Morck et al. (1988) and McConnell and Servaes (1990)). Actually, the empirically estimated threshold value is about 5%, and after this point additional increases in the insider ownership make the relationship between firm value and ownership concentration first decrease and then, (beyond 25%), increase (Morck et al. 1988). These findings are consistent with our results: once a threshold ownership size, necessary to guarantee direct involvement in the management of the firm, is reached, the key issue becomes the trade-off between entrenchment and alignment of interests towards firm value maximization. In such cases, the existence of a second large blockholder monitoring the controlling blockholder may help to reduce expropriation and increasing ownership concentration may have a non-monotone effect on net firm value.

**Funding:** This research was funded by FFR 2023 of University of Palermo.

**Informed Consent Statement:** Not applicable

**Data Availability Statement:** No empirical data were used in this research.

**Conflicts of Interest:** The author declares no conflict of interest.

## Abbreviations

The following abbreviations are used in this manuscript:

| | |
|---|---|
| MDPI | Multidisciplinary Digital Publishing Institute |
| DOAJ | Directory of open access journals |
| TLA | Three letter acronym |
| LD | Linear dichroism |

## Notes

[1] We focus on the pecuniary private benefits of control defined by Gilson (2006) as "the non-proportional flow of real resources from the company to the controlling shareholder". Such benefits broadly include theft, tunnelling, consumption of perks, empire building and in general any use of corporate resources that is not in the best interests of all shareholders.

[2] The basic structure of the model is inspired by Russino et al. (2019).

[3] We can think at the two blocks in terms of a block of insiders, that is a block of shareholders that can directly affect the firm's operations (shareholders being directors, officers or having representatives on the board simultaneously), and a block of outsiders or blockholders that do not have any inside role but can nevertheless affect firm governance gathering information and exerting their voice.

[4] In this paper, we do not investigate how the blocks of shares arise or the process of coalition formation. We assume that there is a controlling block and a non-controlling block and we try to understand how the distribution of ownership between these two blocks affects the ownership–firm value relationship. For the sake of simplicity, from now on we consider the two blocks as individual blockholders.

[5] The restriction on the value of the parameter representing the amount of private benefits of control per unit of company resources diverted is needed to model the more interesting case of inefficient controlling shareholder structures. Assuming $B < 1$ ensures that the extraction of private benefits of control generates a reduction in the company value and implies expropriation of non-controlling shareholders.

[6] Note that $(B - SH_1)$ measures the per unit gain from diversion of company resources realized by the controlling blockholder. Clearly, when the per unit gain is zero there will be no diversion.

[7] Note that the additive functional form used for $D(e, m)$ implies that the cross derivative is equal to zero.

8    The Matlab code used for the simulations is available upon request.

9    That advantage can be related to the fact that the largest shareholder may have an insider position and therefore have a greater ability in gathering information, understanding and influencing the firm's operations and strategic choices.

10    Given two generic twice-differentiable functions $f$ and $g$ on $(L, H)$, the curvature of the two functions is measured by the ratio of the second derivative to the absolute value of the first derivative. The ratios $f''/|f'|$ and $g''/|g'|$ provide a measure of the curvature of the functions, therefore they can be used to determine the relative curvature ordering of the functions: $f$ is more convex than $g$ on $(L, H)$, if $f''/|f'| \geq g''/|g'|$ uniformly on $(L, H)$, or if $f$ is a convex transformation of $g$.

11    As underlined by Nagar et al. (2011), three features characterize closely held corporation: a small number of shareholders; no market for corporate control; involvement of the largest shareholder in the management of the company.

12    For instance, Maury and Pajuste (2005), Laeven and Levine (2008), and Attig et al. (2009) are examples of analyses based on samples of listed firms, while Gutiérrez et al. (2012) and Russino et al. (2019) mainly use samples of unlisted firms.

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
