# Peer review of "Multiple Blockholders and Firm Value: A Simulation Analysis"

_ijfs, doi:10.3390/ijfs11020056_

Round 1

Reviewer 1 Report

Referee report on Multiple Blockholders and Firm Value: A Simulation Analysis

Summary:

This paper lays out a simple model with two blockholders and other additional atomistic investors.  The largest blockholder can divert assets to policies from which they derive private benefits, but there is some cost involved in this diversion.  The second blockholder can also monitor at some cost.  The authors then provide simulations for a variety of parameters, showing that firm value can be more linear or more of an inverted U-curve depending on parameters.

Comments:

I think the paper makes a useful contribution, but it needs to better justify some of the parameters and more clearly present the conclusions.

I think the biggest change that the author should make is to make more effort to justify the parameter choices.  This does not have to be a major calibration, but some discussion of why the simulations examine the given parameters is necessary.  In particular, the author calibrates R to zero in many of the simulations, and this seems like an unrealistic choice.  I do not have strong priors as to what the parameters should be, but I think the paper needs to do a better job justifying why the chosen range of parameters is interesting and relevant.

In the conclusion, I think the author needs to better describe (without relying on mathematical notation) the economic implications of the simulations. 

In terms of literature, I think the author should also tie the findings to papers that examine managerial ownership (similar to the role of blockholdings in this model) and firm value – in particular, Morck, Shleifer, and Vishny (JFE, 1988) or McConnell and Servaes (JFE, 1990).

I also wonder whether the assumption that B<1 is either necessary or realistic.  Consider the classical model presented by Jensen and Meckling (1976), where one manager/owner diverts firm resources to private benefits.  Even if the owner holds 100% of the firm, the owner still chooses some non-zero level of non-pecuniary benefits.  Maybe B really measures the amount of benefits beyond what is chosen if the manager (or SH1 in the current setup) owns 100% of the firm?  Some discussion/clarification of this issue would be helpful. Or the author may want to remove the restriction that there is no diversion if SH1 >= B.

I think the author is really solving for optimal e and m, but the discussion on page 3 makes it sound like they are solving for de/dSH1.  Please clarify.

Also, in the graphs, the lines are difficult to match to the table legend.  Is it possible to get lines that are more distinctive or maybe put fewer lines on each graph?

Author Response

Thank you very much for your insightful and constructive suggestions.

In the revised version of the paper I have highlighted all the changes to make them visible and I added a list of changes at the end of the paper (in the latex file all the changes appear in the blue color).

In this letter, I present the detailed responses to your comments, indicating how and where I have addressed them. To indicate the steps followed in the revision of the paper, I restate your comments below and present point-by-point responses to them.

  • Assumption B<1

Response: I have clarified in the introduction that I focus on “inefficient controlling shareholder structures”. As specified by Gilson (2006)  these are situations where non controlling shareholders are net worse off. On the contrary efficient controlling shareholder structures are those where the benefits of the controlling shareholder monitoring are greater than the cost of private benefits extraction and non controlling shareholders are net better off. In this latter case there is no value loss generated by the extraction of private benefits of control and the only issue may be a distributional one. That is, the increase in firm value due to the managerial activity of the controlling shareholder is not distributed on the basis of the fractional ownership of the shareholders. It is debatable if  there should be any correction: indeed controlling shareholders should receive a remuneration for their involvement in management. On the contrary,  in inefficient controlling shareholder system there is true expropriation of non controlling shareholders  and, therefore, action are needed to prevent it.

I have added in the model section a note specifying this point to justify the restriction B<1.

Additionally, I have added a note explaining that (B-SH1) is the per unit gain from the diversion of corporate resources realized by the controlling blockholder. As a consequence there will be no diversion if the per unit gain is zero or negative.  I have also introduced a note specifying that the model looks at cases where the extraction of private benefits of control requires the use of real company resources (pecuniary private benefits of control). The model does not consider the case of non pecuniary private benefits of control such as social status.

  • Calibration R=0

Response: In the section dedicated to the simulation I have added a part explaining why the case R=0 is an appropriate and interesting starting point. In particular, I explain that this case represent an extreme case of closely held companies but is not unrealistic. I report results of a recent survey on the ownership structure of Italian firms (Baltrunaite et al. 2019) indicating that between 96% and 98% of the ownership is in the hands of the two largest shareholders (R close to zero).

  • Solving for e* and m* or de/dSH1?

Response: In the model section, since I do not specify functional forms,  I do not solve analytically for e* and m*, but using implicit differentiation I can show how the optimal amount of diversion is affected by changes in the ownership distribution (increases in SH1 for given levels of R and B). In the simulations, I specify functional forms and I can, for every specification of the parameters B and R and for different value of SH1, solve for e* and m* and compute the corresponding D*.  

  • Difficult to read the graphs

Response: I have added in the text of the paper and in the captions of the Figures a clarification to match correctly the graph reported in each panel to the legend.

  • More in depth discussion of the economic implications of the simulations with connections to the literature

Response: I have added a discussion in the Conclusions relating the findings of the paper to the literature

  • Relate the findings the managerial ownership literature

Response: In the Conclusions I have also added a paragraph relating the findings discussed in the paper to the literature on managerial ownership.

Reviewer 2 Report

The authors of this study investigate the relationship between ownership concentration and firm value in the presence of multiple blockholders. The article is very interesting, well organized and structured, and the arguments and explanations are clear and convincing. The analyses are provided in an appropriate manner.

However, before it is published, the article may need to be slightly updated. The article is not well anchored in current literature. Therefore, in the Introduction, the literature review part could be expanded with more current research; also, the Conclusions part could be expanded by a more in-depth discussion of problems such as comparisons to the results of prior studies, limitations of the study, among other things.

Author Response

Thank you very much for your insightful and constructive suggestions.

In the revised version of the paper I have highlighted all the changes to make them visible and I added a list of changes at the end of the paper (in the latex file all the changes appear in the blue color).

In this letter, I present the detailed responses to your comments, indicating how and where we have addressed them. To indicate the steps followed in the revision of the paper, I restate your comments below and present point-by-point responses to them.  

  • In the Introduction, the literature review should be expanded with more current research

Response: Following your suggestion In the introduction and in the Conclusions I have added references  strictly related to the topic of the paper and discussed the connections with the results obtained simulating the model.

  • The Conclusions should be expanded introducing comparisons to the results of prior studies, and limitations of the study

Response: In the Conclusions I have added a discussion relating the findings of the paper to the literature. Additionally, I have introduced a paragraph clarifying the limitations of the model. In particular, I stressed that “we acknowledge that we focus on principal–principal conflicts assuming that the familiar managerial agency problem is absent due to the presence of a large blockholder. Our model does not explicitly identify the threshold size of ownership necessary to guarantee convergence of interests between the management and shareholders.” Finally, I have added a discussion of the main empirical evidence on managerial ownership and  related it to our findings.

Reviewer 3 Report

The paper is about exploring the relationship between ownership concentration and corporate resource diverted.  The topic and abstract of the paper are unclear and somewhat misleading.  I thought initially the study was about finding the impact of ownership concentration on firm's value.  The author should revise the title to better reflect the content of the study and explain more clearly on how the distribution of ownership share among controlling block, non-controlling block, and residual ownership would render the corporate resources being diverted.

Author Response

Thank you very much for your insightful and constructive suggestions.

In the revised version of the paper I have highlighted all the changes to make them visible and I added a list of changes at the end of the paper (in the latex file all the changes appear in the blue color).

In this letter, I present the detailed responses to your comments, indicating how and where we have addressed them.  

  • A clarification is needed to explain how changes in the distribution of ownership between the controlling and the non controlling block is related to changes in ownership concentration and how those changes affect firm value.

Response: I have made changes in the manuscript to better explain how, in the model, the distribution of ownership between the controlling block and the non controlling block  can be interpreted in terms of ownership concentration.  

Specifically, I have clarified that, for any given level of diffused ownership (the parameter R in the model), an increase in the fractional ownership of the controlling block implies a reduction in the fractional ownership of the non controlling block. Therefore, for a fixed R, looking at the impact on the optimal total diversion of an increase in SH1 is equivalent to analyze the impact of increasing ownership concentration.  

Additionally, in the section dedicated to the model, I have added an explicit explanation of the link between otpimal diversion as a function of ownership concentration and firm value as a function of ownership concentration. In particular, I have specified that the blockholders security benefits depend on the net firm value (V-D*). The amount of company resources diverted reduces the value on which  blockholders’ fractional ownership is determined.  Thus, if, for instance, D* is a decreasing function of ownership concentration then net firm value will be an increasing function of ownership concentration.